# Vitamin E and Silymarin Reduce Oxidative Tissue Damage during Gentamycin-Induced Nephrotoxicity

**DOI:** 10.3390/ph16101365

**Published:** 2023-09-27

**Authors:** Tsvetelin Georgiev, Galina Nikolova, Viktoriya Dyakova, Yanka Karamalakova, Ekaterina Georgieva, Julian Ananiev, Veselin Ivanov, Petya Hadzhibozheva

**Affiliations:** 1Department of Physiology, Pathophysiology and Pharmacology, Medical Faculty, Trakia University, 6000 Stara Zagora, Bulgaria; tsvetelin.georgiev@trakia-uni.bg (T.G.); viktoriya.dyakova@trakia-uni.bg (V.D.); petya.hadzhibozheva@trakia-uni.bg (P.H.); 2Department of Chemistry and Biochemistry, Medical Faculty, Trakia University, 6000 Stara Zagora, Bulgaria; yanka.karamalakova@trakia-uni.bg; 3Department of General and Clinical Pathology, Medical Faculty, Trakia University, 6000 Stara Zagora, Bulgaria; julian.r.ananiev@trakia-uni.bg; 4Department of Neurology, Psychiatry and Disaster Medicine, Medical Faculty, Trakia University, 6000 Stara Zagora, Bulgaria; veselin.ivanov@trakia-uni.bg

**Keywords:** ferroptosis, GPX4, ROS, lipid peroxidation, gentamicin

## Abstract

Aminoglycoside antibiotics and gentamicin (GN), in particular, are still widely used in clinical practice. It is a well-known fact that GN causes nephrotoxicity, and redox disturbances are discussed as a factor in its side effects. Recently, a new type of cell oxidative death, named ferroptosis, was discovered; it is associated with iron accumulation in the cell, glutathione (GSH) depletion and inactivation of glutathione peroxidase-4 (GPX4), reactive oxygen species (ROS) increment with concomitant lipid peroxidation. In this regard, a possible connection between GN-induced renal damage, ferroptosis and the overall antioxidant status of the organism could be investigated. Moreover, due to its beneficial effects, GN is still one of the main choices as a therapeutic agent for several diseases, and the possible reduction of its side effects with the application of certain antioxidants will be of important clinical significance. The study was conducted with adult male white mice divided into several groups (n = 6). GN nephrotoxicity was induced by the administration of GN 100–200 mg/kg i.p. for 10 days. The control group received only saline. The other groups received either Vitamin E (400 mg/kg p.o.) or Silymarin (200 mg/kg p.o.) applied alone or together with GN for the same period. After the end of the study, the animals were sacrificed, and blood and tissue samples were taken for the assessment of biochemical parameters and antioxidant status, as well as routine and specific for GPX4 histochemistry examination. The experimental results indicate that GN-induced nephrotoxicity negatively modulates GPX4 activity and is associated with increased production of ROS and lipid peroxidation. The groups treated with antioxidants demonstrated preserved antioxidant status and better GPX4 activity. In conclusion, the inhibition of ROS production and especially the suppression of ferroptosis, could be of clinical potential and can be applied as a means of reducing the toxic effects of GN application.

## 1. Introduction

Ferroptosis is a type of cell death formally introduced as a concept in 2012 that is associated with cellular iron accumulation and lipid peroxidation [1]. The pathways by which ferroptosis leads to cell destruction are different than those that induce apoptosis. In ferroptosis, the leading mechanism is the inactivation of Glutathione peroxidase-4 (GPX4), increased redox potential, ROS accumulation and subsequent lipid peroxidation, as well as oxidative cell death [2,3]. Other significant changes in ferroptosis are associated with increased expression levels of acyl-CoA synthetase long-chain family member 4 [4], while NADPH-FSP1-CoQ10 pathway and GTP cyclohydrolase 1 (GCH1), a rate-limiting enzyme in the BH4 biosynthesis pathway, are important for the suppression of the ferroptosis as well [5,6]. Ferroptosis plays an important role in the pathophysiology of several disorders, such as tumor development, neurological diseases, vascular problems, and renal failure [2,7,8]. The significant role of ferroptosis in studies of some models of kidney damage (like folic acid-induced damage) has been reported, and the inhibition of the process almost completely neutralized the pathological changes in the kidneys [9]. In recent years, the mechanisms of ferroptosis, as well as the factors that affect it, are beginning to be revealed, but many research efforts are still needed to study this process well enough. Zheng and coworkers [10] showed that ferroptosis plays an important role in neomycin-induced ototoxicity. It is a well-known fact that in addition to ototoxicity, the aminoglycoside antibiotics cause nephrotoxicity as well. Therefore, it is of great interest to study the relationship between aminoglycoside (and gentamicin in particular)-induced renal damage and ferroptosis. 

Moreover, due to its beneficial effects, gentamicin (GN) is still one of the main choices as a therapeutic agent for several infections, and the possible reduction of its side effects by the application of certain compounds will be of important clinical significance [11]. In addition to the newly synthesized specific inhibitors of ferroptosis, the so-called ferrostatins such as Ferrostatin-1, Liproxstatin-1, and SRS 16-86 exist, and natural ones include Vitamin E [12]. The search for other natural antioxidants with potential ferrostatic effects like silymarin (SM)—a bioactive extract from the plant *Silybum marianum* L. that contains flavonolignans like silybinin [13], will expand their clinical potential for ferroptosis-related known disorders. The potent antioxidant [14] and hepatoprotective effects [15] of silymarin, as well as its nephroprotective properties the extract [16,17], are well known. In the present study, we aimed to investigate the in vivo protective properties of vitamin E and SM against GN-induced nephrotoxicity and their relevance to accompanying oxidative changes. Our hypothesis is that ferrostatins have clinical potential and can be applied as a means of reducing the toxic effects of aminoglycoside treatment.

## 2. Results

### 2.1. Physiological and Biochemical Status

At the end of the experimental period, the mice were sacrificed, and blood samples were collected for biochemical analysis. Table 1 shows the results of creatinine, urea, and electrolytes Na^+^ and K^+^. Significant changes were observed only in the level of blood urea and creatinine in the GN-treated group. The level of urea in the combined groups was lower than the GN-treated group, *p* < 0.05.

### 2.2. Kidney Histopathology

Mild degenerative and inflammatory changes were observed in the kidney tubules of the gentamicin-treated group. In the controls and the other groups, including combined groups, there are almost no visible changes. Only in the gentamicin plus silymarin-treated group did more pronounced vascular congestion still exist. Regarding GPX4 expression, the results were more variable (Figure 1). There was no significant difference between the control group and the gentamicin-alone-treated group, where GPX4 expression was strong. In silymarin and gentamicin plus silymarin-treated groups, the GPX4 expression was moderate. The biggest diversity was demonstrated where the GPX4 expression was weak with the mice treated only with Vitamin E, but the combination of gentamicin plus vitamin E led to very strong GX4 expression.

The kidney tissue examination from the six groups of experimental animals showed varying degrees of morphological changes. We used our own protocol to account for pathomorphological changes in different groups according to their severity (Table 2).

Levels of GPX4 measured in the serum of mice are shown in Figure 2. The level of GPX4 in the gentamicin-treated group was statistically higher compared with the control one (1623.3 ± 138.2 pg/mL vs. 1280.7 ± 112.3 pg/mL, *p* < 0.05. The silymarin-treated group demonstrated the highest GPX4 levels (2559.0 ± 186.8 pg/mL). Other experimental groups showed similar levels (1007.8 ± 78.1 pg/mL; 1131.5 ± 81.3 pg/mL for the combined groups; and 1060.8 ± 83.2 pg/mL for the vitamin E group, respectively) as the control one (*p* > 0.05), but statistically significant with the gentamicin-only-treated group *p* < 0.05.

### 2.3. Analysis of Hydroxyproline (Hyp), MDA and 8-OHdG in Kidney Tissue

Figure 3 presents the GN toxicity by measuring the hydroxyproline content in kidney tissue and lipid peroxidation levels measured as MDA and DNA oxidation measured as 8-OHdG substances.

GN administration statistically significantly increased the Hyp levels vs. controls (849.2 ± 63.3 mg/g tissue vs. 425.7 ± 69.8 mg/g tissue, *p* < 0.05). The results presented in Figure 3A show that GN-induced kidney damage was compensated by vitamin E or silymarin protection. The Hyp level was statistically significantly reduced in the group treated with the combination of vitamin E and GN (498.2 ± 49.0 mg/g tissue, *p* < 0.05), as well as in the group with the combination of silymarin and GN (491.0 ± 55.3 mg/g tissue, *p* < 0.05). The groups protected for 10 days with antioxidants only showed lower levels with no statistically significant difference compared to the controls (Figure 3A). In the GN-treated mice, the MDA levels were significantly increased in the kidney compared to the controls (5.97 ± 0.31 µmol/mL vs. 3.09 ± 0.18 µmol/mL, *p* < 0.05, Figure 3B). Pretreatment with vitamin E or silymarin effectively brought down kidney MDA levels compared to the GN-treated group and close to the control (4.01 ± 0.14 µmol/mL and 3.98 ± 0.16 µmol/mL, respectively, *p* < 0.05, Figure 3B). The same tendency was observed regarding the levels of 8-OHdG, where the kidney tissue from the GN-treated group was with significantly increased levels of 8-OHdG compared to the controls (6.85 ± 0.37 ng/mL vs. 3.55 ± 0.23 ng/mL, *p* < 0.05, Figure 3C). In the group treated with the combination of vitamin E and GN as well as silymarin and GN, the 8-OHdG level was statistically significantly reduced (4.01 ± 0.34 ng/mL and 3.96 ± 0.23 ng/mL, respectively, *p* < 0.05, Figure 3C).

### 2.4. Activity of Antioxidant Enzymes

GST activity (Figure 4A) was statistically significantly increased in the kidney of GN injured mice compared to the control (892.1 ± 33.9 nmol/gPr vs. 512.4 ± 74.9 nmol/gPr, *p* < 0.05) and the protected with antioxidants groups (625.1 ± 41.5 nmol/gPr and 517.4 ± 42.5 nmol/gPr, respectively, *p* < 0.05). The groups treated with a combination of vitamin E and GN, as well as silymarin and GN, showed a statistically significant decrement in GST activity compared to the GN group (628.5 ± 32.6 nmol/gPr and 618.0 ± 33.8 nmol/gPr, respectively, *p* < 0.05, Figure 4A). The SOD activity (Figure 4B) in the GN group was significantly decreased compared to the control (1.29 ± 0.17 U/gPr vs. 4.74 ± 0.13 U/gPr, *p* < 0.05), whereas in the other four groups (only antioxidants and with combination), the levels of SOD activity were similar to the controls (3.81 ± 0.18 U/gPr and 5.01 ± 0.36 U/gPr, respectively, for the antioxidants alone, *p* < 0.05) and (3.98 ± 0.22 U/gPr and 4.23 ± 0.21 U/gPr, respectively, for the combinations, *p* < 0.05), Figure 4B).

Statistically significant lower GSPx kidney activity was measured in mice treated with GN compared to the healthy controls (19.0 ± 2.3 U/gPr vs. 68.8 ± 3.5 U/gPr, *p* < 0.05, Figure 4C), and in the groups treated with a combination of vitamin E and GN (50.8 ± 3.7 U/gPr, *p* < 0.05) as well as with a combination of silymarin and GN (61.2 ± 4.8 U/gPr, *p* < 0.05). The kidney GSPx activity in antioxidants alone mice were significantly elevated compared to the GN group (48.0 ± 3.5 U/gPr and 69.1 ± 4.6 U/gPr, respectively, *p* < 0.05).

### 2.5. Determination of Oxidative Protein Remodeling in Kidney Tissue

The AGEs levels (Figure 5A) were statistically significantly increased in the GN-treated group compared to the control group (852.6 ± 69.5 mg/mL vs. 243.4 ± 85.2 mg/mL, *p* < 0.05). A statistically significant increase compared to the controls was observed in the groups protected with antioxidants only, especially in the vitamin E-only-treated group (609.7 ± 71.3 mg/mL and 304.0 ± 58.9 mg/mL, respectively, *p* < 0.05). A statistically significant reduction compared to the GN group in AGEs levels was detected in the groups with the combination of vitamin E and GN and silymarin and GN (531.7 ± 36.7 mg/mL and 489.6 ± 33.9 mg/mL, respectively, *p* < 0.05).

The PCC levels (Figure 5B) in the group treated with GN were statistically significantly increased compared to the controls (11.21 ± 0.29 nmol/mg vs. 5.02 ± 0.12 nmol/mg, *p* < 0.05). No significant changes in PCC levels were indicated between the mice treated with antioxidants alone and the control group (6.88 ± 0.26 nmol/mg and 5.59 ± 0.33 nmol/mg, respectively, *p* > 0.05). A statistically significant difference was observed in the GN-treated group compared to the protected groups with the combination of vitamin E and GN as well as silymarin and GN (6.01 ± 0.24 nmol/mg and 5.74 ± 0.14 nmol/mg, respectively, *p* < 0.05).

### 2.6. Parameters of Oxidative Damage in Kidney Tissue and Serum

The statistically significant increase in the NO radical levels (Figure 6A,D), ROS levels (Figure 6B,E) and ascorbate (Asc) radical levels (Figure 6C,F) compared to the controls were measured in the kidney and serum, respectively (*p* < 0.05). A statistically significant reduction in NO radical level was measured in the group treated with the combination GN +Vit E group (mean 18.65 ± 2.15 vs. 55.61 ± 3.13, *p* < 0.05) and GN + SM in kidney tissue (mean 17.54 ± 3.22 vs. 55.61 ± 3.13, *p* < 0.05) compared to the GN-treated group and healthy controls (Figure 6A). In the serum samples, the NO was also statistically significantly reduced in groups treated with protector in combination with GN compared to the GN group (mean 19.34 ± 2.33 vs. 41.54 ± 3.25 Vit E +GN, *p* < 0.05) and (mean 17.51 ± 3,08 vs. 41,54 ± 3,25 SM + GM, *p* < 0.05). The NO levels in both treated groups were close to the control group (*p* > 0.05).

The ROS levels (Figure 6B) were statistically significantly elevated in the GN-treated group compared to the control (mean 2.77 ± 0.08 vs. 0.89 ± 0.05) and both protected groups measured in kidney (mean 2.77 ± 0.08 vs. 1.11 ± 0.4 (Vit E + GN) and mean 2.77 ± 0.08 vs. 1.23 ± 0.3 (SM + GM) *p* < 0.05). In the serum ROS levels in protected groups, Vit E + GN and SM + GN were statistically significantly lower compared to the GN-treated alone (*p* < 0.05) and close to the control group (Figure 6E).

Ascorbate radicals levels measured in the GN-treated group were statistically significantly elevated in the kidneys compared to protected groups (mean 4.88 ± 0.36 vs. 2.39 ± 0.39, Vit E + GN, and mean 4.88 ± 0.36 vs. 2.61 ± 0.58, SM + GN, *p* < 0.05) and to the controls (mean 4.88 ± 0.36 vs. 1.31 ± 0.41 *p* < 0.05) (Figure 6 C). The same was observed in Asc radical levels measured in the serum of the GN-treated group compared to the controls *p* < 0.05 Figure 6F. In protected groups with Vitamin E and SM, the GN the Asc levels were close to the controls.

### 2.7. IL-1b, IL-6, IL-10 in Kidney Homogenate/Serum

Compared to the controls, GN treatment increased the IL-1β levels in the kidney homogenate and serum (177.1 ± 25.5 pg/mL vs. 144.7 ± 25.5 pg/mL, *p* < 0.05) and (174.3 ± 35.9 pg/mL vs. 140.5 ± 31.6 pg/mL, *p* < 0.05). IL-6 levels shown the same tendency (181.3 ± 39.9 pg/mL vs. 145.2 ± 40.8 pg/mL; *p* < 0.05) and (181.2 ± 50.8 pg/mL vs. 142.8 ± 39.6 pg/mL, respectively; *p* < 0.05); however, the IL-10 levels were also increased in the serum (241.2 ± 70.1 pg/mL vs. 186.4 ± 59.0 pg/mL; *p* < 0.05), but significantly decreased in the kidney (4.80 ± 0.05 pg/mL vs. 7.10 ± 0.05 pg/mL; *p* < 0.05), (Figure 7). IL-1β and IL-6 (kidney and serum) and IL-10 levels (serum) in the groups protected with vitamin E or silymarin alone and the combination of these antioxidants and GN showed results close to the controls (*p* > 0.05). In IL-10 kidney levels, the vitamin-E-only protected group showed an insignificant tendency for increase compared to the GN-treated (*p* > 0.05).

### 2.8. TNF-a and INF-γ Concentration

GN administration increased the concentrations of TNF-α (15.30 ± 0.05 pg/mL vs. 9.01 ± 0.08 pg/mL) in the kidney homogenate and in the serum compared to the controls (82.8 ± 12.9 pg/mL vs. 63.2 ± 22.0 pg/mL). The same tendency was observed regarding INF-γ where in the kidneys, its levels were (12.52 ± 0.06 pg/mL vs. 8.20 ± 0.05, *p* < 0.05) and in the serum (181.3 ± 23.5 pg/mL vs. 145.5 ± 11.1 pg/mL, *p* < 0.05) (Figure 8). TNF-α and INF-γ (kidney and serum) in the groups protected with vitamin E or silymarin alone and the combination of these antioxidants and GN showed results close to the controls (*p* > 0.05) but significantly lower compared to the GN-induced toxicity group (*p* < 0.05), respectively.

### 2.9. EPR Measurement of Oxidative Stress by 4-Hydroxy-2,2,6,6-tetramethylpiperidine 1-Oxyl (TEMPOL)

Electron paramagnetic resonance is a non-invasive, highly sensitive technique for studying free-radical processes both “in vitro” and “in vivo”. EPR is considered the only method that can be used to directly monitor ROS in complex biological systems [18]. Free radicals are usually very reactive and have short half-lives. The detection of an equilibrium state in which their concentration is below the EPR detection limit necessitates the use of spin probes that improve the contrast through electron transfer reactions to a diamagnetic form of the nitroxides–hydroxylamine [19]. In this context, nitroxide radicals are the most widely used contrast markers in nitroxide-enhanced EPR spectroscopy. X-band EPR spectroscopy, with the participation of nitroxides, occupies one of the most important places among spectroscopic techniques in the study of redox changes in cells, tissues, and organs [20].

GN treatment induced high levels of free radicals, and as a result, the EPR signal intensity and the double-integrated area of the nitroxide spectrum decreased dramatically compared to the control (mean 19.7 ± 2.39 vs. 80 ± 5.1, *p* < 0.05). The results show (Figure 9) that vitamin E and Silymarin show a protective effect on gentamicin-induced oxidative stress (Vit E + GN 56.7 ± 1.8 vs. 19.7 ± 2.39 a.u., *p* < 0.05), and (SM + GN 69.8 ± 2.1 vs. 19.7 ± 2.39 a.u., *p* < 0.05). Silymarin as a protector shows a higher potential effect compared to traditional antioxidant, Vitamin E.

## 3. Discussion

The present work aimed at establishing a GN model of nephrotoxicity and to study the possible relationship between aminoglycoside (and GN in particular)-induced kidney damage and ferroptosis. Moreover, in this research, we investigated the in vivo protective properties of vitamin E and SM against GN-induced nephrotoxicity and their relevance to accompanying oxidative and morphological renal changes. This could be of important clinical significance in the efforts to reduce the toxic effects of aminoglycoside treatment.

The biochemical (urea and creatinine levels) and histopathological results demonstrate that the administration of 200 mg/kg GN for 10 days in mice leads to moderate nephrotoxicity. Lower dosages (50 or 100 mg/kg) or shorter periods (5 or 7 days) are almost without histochemical and biochemical consequences. It has been reported that the nephrotoxicity induced by GN involves different pathways, including oxidative stress (OS), inflammation, nitric oxide (NO) generation, lipid peroxidation, and decreased efficiency of kidney antioxidant enzymes such as superoxide dismutase (SOD), catalase, glutathione peroxidase, and reduced glutathione (GSH) levels [21]. This is in accordance with the results obtained in our study, which revealed that the GN-treated group had increased (OS) by depleting the activity of antioxidant enzymes and showing a drastic increase in ROS/RNS levels, leading to protein damage and increased levels of inflammatory cytokines. Oxidative imbalance induces damage to the structure of nucleic acids, lipids, and proteins and impairs the synthesis of prostaglandins, leukotrienes, and thromboxanes [22].

Furthermore, due to the elevation of inflammatory cytokines, the oxidative homeostasis in the nephrotoxicity-induced group was additionally disturbed [23]. Damaged proteins and lipids accumulated in stressed cells alter the structure and function of proteins and stimulate both NADPH-dependent and ascorbate-dependent lipid peroxidation with co-factor ferrous ions (Fe^3+^) [22]. Via the cytochrome P450 reductase system, Fe^3+^ generates •OH, thus increasing the MDA and ROS levels [24]. Excessive ROS overproduction (especially in the context of ferroptosis and lipid damage) leads to membrane disruption and ion imbalances [25]. The presented results in our study showed significant changes in 8-OHdG in the GN-treated group, suggesting oxidative nuclear and mitochondrial DNA damage [26].

The affected interleukins, together with the change in the expression of antioxidant enzymes (SOD and glutathione peroxidase) and increased Fe^3+^ and ROS production, suggest GSH and Sirtuin 1 (SIRT1) depletion, which could be the connection between OS, mitochondrial damage and ferroptosis [25,27]. GSH is the most important antioxidant synthesized in cells and helps to remove peroxides and many xenobiotic compounds [28]. In addition, GSH is a cofactor and substrate for glutaredoxins (Grx), which catalyze disulfide reductions in the presence of NADPH and glutathione reductase (GR). Protein and lipid damage is prevented by enzymes such as GR, multiple Grx and glutathione-S-transferases (GST). The fact that in the GN-treated group, the GST levels were almost twofold higher than the controls, but SOD and GSPx were three times less, is another sign of GSH depletion, nephrotoxicity, and the activation of ferroptosis. Glutathione peroxidases (GPXs) are differentially expressed in cells and tissues and are involved in H_2_O_2_ detoxification. GPX4 reduces the cytotoxic lipid peroxides (L-OOH) to the corresponding alcohols (L-OH), and its activity is the main pathway of ferroptosis [29]. Different pathways for the inhibition or initiation of ferroptosis exist, but a central role in the process plays GSH and GPX4 bioavailability and activity [30]. In our study, the relatively low change in the kidney expression of GPX4 in the GN-treated group and the control one supports the indirect inhibitory GPX4 mechanism for ferroptosis induction, which has been reported previously by some authors [31]. However, an additional direct inhibition of GSH/GPX4, followed by a subsequent compensatory reaction of the antioxidant-protective body system, could be assumed, too. Such reactions might affect the nuclear factor erythroid 2 (Nrf2), which is an important regulator of the ferroptosis [32]. This suggestion corresponds with the results of Yue and coworkers [33] in the knockout Nrf2 rat model of nephrotoxicity.

The recovery of the biomarkers of oxidative stress through the reduction of oxidation products of proteins and lipids (AEGs, PCC), stimulation of the activity of antioxidant enzymes, and suppression of ROS and RNS formation in the kidney and serum is evidence for the protective effect of vitamin E and SM against GN-induced nephrotoxicity. Moreover, the groups treated with these antioxidants showed protection from kidney inflammations, and the results were similar to the controls. Vitamin E possesses a strong protective effect as a lipophilic antioxidant and has the ability to inhibit the formation and propagation of oxidized lipids and suppress GSH depletion [29]. Due to its role in the GSH/GPX4 axis, vitamin E is considered a potent natural ferrostatin. Many studies confirm its protective role against OS and kidney injuries [34,35]. Together with other parameters of OS and inflammation that were very low in the vitamin E-treated group, the registered weak expression of GPX4 in the kidneys of these mice is probably a sign of a down-regulation of the enzyme due to the better oxidative status. This is in accordance with Sneddon et al. [32], who found that α-tocopherol (the most active form of vitamin E) reduced mRNA GXP4 levels without affecting its activity.

In contrast to the vitamin E-treated group, the one with SM showed twofold higher levels of GPX4 in the serum. Such an increase might be because SM could prevent OS and eventually ferroptosis by several mechanisms, including direct scavenging of free radicals, chelating free Fe and Cu, inhibiting specific ROS-producing enzymes, improving mitochondrial integrity, activating Nrf2 pathway, inhibiting NF-κB pathways, activating heat shock proteins, etc. [14]. The obtained results demonstrated that natural ferrostatins like the powerful vitamin E, as well as the new potential active antioxidant SM, regulate the OS redox-homeostatic imbalance and the immunomodulatory response after GN-induced nephrotoxicity in experimental mouse models and benefit the restoration of kidney function. Gentamicin and the other aminoglycosides affect bacterial ribosomes, leading to the formation of non-functional proteins, as well as ROS production, because of this protein disturbance [36]. Since OS is part of the GN’s antibacterial activity, any antioxidant supplementation could affect GN’s effectiveness. However, GN-associated oto- and nephrotoxicity is mainly due to OS. In this regard, the benefits of ferrostatins could be discussed when longer GN treatment is necessary, especially in polymorbid patients, where nephrotoxicity is associated with higher mortality [37]. In many aspects, the application of SM at a dose of 200 mg/kg p.o. showed similar or even better protection than the vitamin E administered at a dose of 400 mg/kg p.o. (IL-10, AEGs, SOD) in the combined groups. Additionally, to its protective antioxidant properties, in the GN + SM combined group, no inflammatory changes in the kidney were observed, and the degenerative process was diminished. The ability of SM to activate the SIRT1 pathway and to act as an indirect activator of the Nrf2 signaling pathway that is involved in mitochondrial function and survival has been proven in other studies [14,38]. Furthermore, the administration of vitamin E and SM in combination with GN treatment restored the AEGs and especially PCC levels almost to the control levels.

The protective effect of Vit E and SM against gentamicin-induced nephrotoxicity in mice kidney tissue and serum was evaluated by pro- and anti-inflammatory markers. The GM-treated leads to an inflammatory reaction expressed in an increase in the pro-inflammatory markers IL-1β and IL-6, while the anti-inflammatory IL-10 is statistically significantly decreased compared to the control and protected groups. The results showed that Vitamin E and Silymarin reduced gentamicin-induced toxicity in kidney tissue (Figure 7A–C). The serum level of IL-10 was significantly increased, while IL-6 and IL-1β were decreased in the gentamicin group compared with the protected groups (*p* < 0.05) (Figure 7D–F).

In the GN-induced nephrotoxicity model, ROS/RNS generation resulted in progressive lipid peroxidation that had an impact on the reductive protein metabolism, increased collagen synthesis by increasing hydroxyproline, and directly affected the antioxidant defense system [25]. The anti-inflammatory cytokine IL-10 increases antibody production as well [39,40]. IL-10 functions by inhibiting proinflammatory cytokines produced by macrophages and regulatory T cells, including IFN-gamma and TNF-alpha. In the present study, vitamin E and even more SM, were shown to significantly inhibit GN-induced levels of IL-1β, IL-6 (Figure 7), TNF-α, and IFN-gamma (Figure 8) in mice and to increase IL-10 levels. Our findings correspond with those presented by Akbaribazm et al. [41], which demonstrated that herbal antioxidants have strong protective effects against gentamicin-induced nephrotoxicity and modulate anti-inflammatory markers in cases of renal diseases.

Nitroxides are a class of piperidine (TEMPOL, TEMPO, Mito-TEMPO, etc.) and pyrrolidine free radicals (Carboxy-PROXYL, Carbamoyl-PROXYL, etc.) used in CW-EPR, SDSL-EPR and MRI techniques. They are redox-sensitive agents that can be reduced to the corresponding diamagnetic forms—hydroxylamines or oxoammonium cations by one-electron transfer reactions [42,43]. Like endogenous SOD, nitroxide radicals act as catalysts in the dismutation process of O_2_•^−^ to H_2_O_2_ and oxygen and can accumulate intracellularly at concentrations that effectively reduce superoxide anion radicals. The rate-determining step of this process is the oxidation of radical form to hydroxylamine. Studies have shown that the reverse process of oxoammonium cation reduction to O_2_•^−^ radical occurs at a higher rate than the corresponding SOD reaction [44,45]. Piperidine nitroxides TEMPOL is a membrane-permeable nitroxide that has pronounced SOD mimetic activity [46,47]. Experimental studies have shown that nitroxide ameliorates oxidative stress-mediated renal dysfunction and glomerular injury [48], ameliorates endothelial cell dysfunction in diabetic rats and reduces the infarct size of regional myocardial ischemia/reperfusion [49].

Cyclic nitroxide radicals are defined as suitable spin detectors for assessing the degree of oxidative stress in various pathologies. Due to their ability to interact with free radicals, they are widely used in the development of many experimental models for monitoring OS global cellular redox status [50]. The results presented in Figure 9 summarize the experimental protocol aimed at studying changes in the TEMPOL EPR signal intensity (a.u.) in mouse kidney tissue samples and comparing the obtained results with the intensity of the EPR spectrum in a pure TEMPOL/DMSO system (control sample, without tissue).

## 4. Materials and Methods

### 4.1. Plant Material and Drugs

Silymarin was collected from central Bulgaria at the end of 2022. All chemicals were analytical grade (AR > 99.7%). The used silymarin was donated by Prof. V Ivanov, Trakia University, Stara Zagora, Bulgaria [17]. Vitamin E was purchased by Sigma Aldrich Pty Ltd., Merck KGaA, Sofia, Bulgaria. Gentamicin was purchased in the pharmacy forms.

### 4.2. Animals, Experimental Design, and Ethical Approval

White male mice (n = 36) aged 6 weeks (mean weight 27–36 ± 2.0 g) were purchased from the Institute of Animal Science, Slivnitsa, Bulgaria. After transport, the animals had a 10-day cycle of adaptive feeding and acclimatization. During the experiment, they had ad libitum access to fresh water and food, maintained 12 h/d light/dark cycles, 23 ± 2 °C and humidity 55% in the vivarium of the Faculty of Medicine of Trakia University. The work procedures were approved by the Trakia University Animal Ethics Commission, Stara Zagora, Bulgaria, and the Bulgarian Food Safety Agency, Sofia, Bulgaria, with a license (325/6000-0333/09.12.2021) following Directive 2010/63/EU on the animals’ protection used for experimental and other scientific work. The experimental animals were divided into six groups (n = 6) under controlled environmental conditions: (1) control group (n = 6); (2) gentamicin-induced nephrotoxicity group (n = 6) treated by GN for 10 consecutive days (200 mg/kg i.p.); (3) vitamin E group (n = 6) treated by vitamin E in a dose of 400 mg/kg p.o., given via a feeding needle for 10 consecutive days; (4) silymarin group (n = 6) treated by silymarin in a dose of 200 mg/kg p.o., given via a feeding needle for 10 consecutive days; (5) vitamin E and GN group (n = 6) treated by vitamin E in a dose of 400 mg/kg p.o., given via a feeding needle for 10 consecutive days and GN for 10 consecutive days (200 mg/kg i.p.); (6) silymarin and GN group (n = 6) treated by silymarin in a dose of 200 mg/kg p.o., given via a feeding needle for 10 consecutive days and GN for 10 consecutive days (200 mg/kg i.p.). The used vitamin E and silymarin were mixed with d.H_2_O and crude olive oil (Lekkas Farm, Mikro Horio, Greece) before treating the mice. The physiological state and behavior of the experimental animals were monitored daily.

### 4.3. Dissection Procedure

The animals were sacrificed under anesthesia (Nembutal 50 mg/kg i.p.) on day 11 from the beginning of the experiment. Blood samples were collected by intracardiac standard technique, and fresh blood was collected with vacutainer serum tubes. Serum samples were prepared by centrifugation (4000 rpm, 10 min, 4 °C). Kidney samples were stored in ice-cold PBS (4 °C), weighed (SW) and homogenized.

### 4.4. Histological Analysis for Visualization of Kidney Changes

The preparation of the kidney samples for histopathological examination included removal and perfusing of the right kidneys, followed by immersing the tissues in a fixative 10% aqueous formalin solution for 24 h. After dehydration in a graduated series of ethanolic concentrations, the blocks are clarified in xylene and embedded in paraffin. The tissue 4 µm-sections were mounted on gelatin-coated slides, dewaxed twice in xylene and rehydrated by a series of decreasing ethanol concentrations. The histological evaluation was performed after staining the sections with a standard hematoxylin/eosin-based method (0.1% H&E) and specific staining with antibodies against GPX4 (Glutathione Peroxidase 4/GPX4 Antibody (E-12), Santa Cruz Biotechnology, Inc., Dallas, TX, USA).

### 4.5. Analysis of Hydroxyproline (Hyp) in Kidney Tissue

The analysis for Hyp levels in the kidneys was used to measure tissue collagen content indirectly and to quantify the tissue damage by a Woessner method [51]. In brief, analysis of hydroxyproline levels in kidney tissue (drying at 110 °C for 24 h; hydrolysis with 6N HCl, incubation at 110 °C) was determined spectrophotometrically by 550 nm absorption, and the results are expressed as μg of hydroxyproline per gram used tissue.

### 4.6. Determination of Glutathione S-Transferase (GST)

The glutathione-S-transferase activity (GST) in kidney homogenates was determined using a 1-chloro-2,4-dinitrobenzene reagent [52]. The chemical mixture included 0.1 M phosphate buffer (pH = 8.2), 15 mM GSH, and 15 mM CDNB, which were added to a 10 µL tissue homogenate sample. A control sample contained only a chemical reaction mixture without tissue homogenate. The enzyme production and, respectively, the activity of GST in each investigated sample were evaluated through the spectrophotometric sample investigation with absorption at 340 nm.

### 4.7. Protocol for Functional Markers Measurement of Kidney Damage

For the evaluation of the functional kidney damage, clinical data were analyzed for creatine, urea and electrolytes. Creatinine and urea were measured with ELISA kits (Abcam, Cambridge, UK) by following the manufacturer’s instructions, and electrolytes were measured by an electrolytic panel.

### 4.8. Electron Paramagnetic Resonance (EPR) Study

All EPR measurements were performed at room temperature on a Bruker BioSpin GmbH, Ettlingen, Germany, equipped with a standard resonator. All EPR experiments were carried out in triplicate and repeated thrice. Spectral processing was performed using Bruker WIN-EPR and Simfonia software, version 2015.

#### 4.8.1. Evaluation of the ROS Product Levels

The levels of ROS were determined following Shi et al. [53] with some modifications. To investigate the ROS formation in the samples, ex vivo EPR spectroscopy was used combined with N-tert-butyl-alpha-phenylnitrone (PBN) as a spin-trapping agent. Briefly, to 100 μL plasma was added 900 μL 50 mM PBN dissolved in dimethyl sulfoxide (DMSO), and, after centrifugation at 4000× *g* rpm for 10 min at 4 °C, the EPR spectra were immediately recorded in the supernatant. The levels of ROS products were calculated as double-integrated plots of EPR spectra, and results were expressed in arbitrary units (a.u.). The EPR settings were as follows: 3503.73 G center field, 20.00 mW microwave power, 5 G modulation amplitude, 50 G sweep width, 1 × 10^5^ gain, 81.92 ms time constant, 125.95 s sweep time, and five scans per sample.

#### 4.8.2. Evaluation of the •NO Radical Levels

Based on the methods published by Yoshioka et al. [54] and Yokoyama et al. [55], we developed and adapted the EPR method for the estimation of •NO radical levels. Briefly, to a 50 μM solution of carboxy 2-(4-carboxyphenyl)-4,4,5,5-tetramethyl was added imidazoline-1-oxyl-3-oxide potassium salt (carboxy PTIO.K) dissolved in a mixture of 50 mMTris (pH 7.5) and DMSO in a ratio 9:1. To the 100 μL samples was added 900 μL Tris buffer dissolved in DMSO (9:1). After that, the mixture was centrifuged at 4000× *g* rpm for 10 min at 4 °C. Then, 100 μL of the sample and 100 μL 50 mM solution of carboxy PTIO.K were mixed and EPR spectra of spin adducts formed between the spin trap carboxy PTIO.K and generated •NO radicals were recorded. The •NO radicals levels were calculated as double-integrated plots of EPR spectra, and the results were expressed in a.u. The EPR settings were as follows: 3505 G centerfield, 6.42 mW microwave power, 5 G modulation amplitude, 75 G sweep width, 2.5 × 10^2^ gain, 40.96 ms time constant, 60.42 s sweep time, and one scan per sample.

#### 4.8.3. TEMPOL

A nitroxide radical solution (50 µL 2 mM) was added to the kidney homogenate samples and stirred on a vortex for 5 s at room temperature. The reaction mixture (nitroxide/serum sample) is incubated for 10 min. The samples are taken in a microcapillary, placed in the EPR cavity, and started measurements. Each sample was scanned twice and repeated [56].

### 4.9. Immunoenzyme Assays

#### 4.9.1. Determination of the Activity of Antioxidant Enzyme System and Products of Oxidation of Proteins and Lipids in the Kidney

To determine the enzyme activity of superoxide dismutase (SOD) and glutathione peroxidase (GSPx), and oxidative stress parameters malondialdehyde (MDA), protein carbonyl content (PCC) and glycation end products (AGEs) were used with ELISA kits (Wuhan Fine Biotech Co., Ltd., Wuhan, China) by following the manufacturer’s instructions.

#### 4.9.2. Measurement of Proinflammatory Markers in Kidney Tissue and Blood

The cytokine levels (IL-1β, IL-10, IL-6, ITF-γ, and TNF-α) were measured with ELISA kits at the end of the experiment.

### 4.10. Statistical Analysis

The obtained results were processed with Statistica 8.0 (StatSoft, Inc., Tulsa, OK, USA) and presented as mean ± standard error (SE). Statistical analysis was performed using one-way ANOVA and a post-hoc LSD test to determine significant differences among data groups. A *p*-value < 0.05 was considered to be statistically significant.

## 5. Conclusions

In the present study, we reported significantly elevated oxidative stress with accompanying increased production of ROS, decreased redox potential, increased production of cytokines, and evidence for changes in the GPX4 axis and ferroptosis in the GN-treated mice. Protection with vitamin E (400 mg/kg p.o.) or silymarin (200 mg/kg p.o.) recovers most of the parameters of inflammation, oxidative stress and ferroptosis. The results presented above suggest that inhibition of GN-induced kidney injury affects not only the survival of the cells but also improves their anti-inflammatory status and oxidative balance. Vitamin E and silymarin as ferrostatins could be used as complementary agents against gentamicin-induced side effects and, in particular, its nephrotoxicity.

## Figures and Tables

**Figure 1 pharmaceuticals-16-01365-f001:**
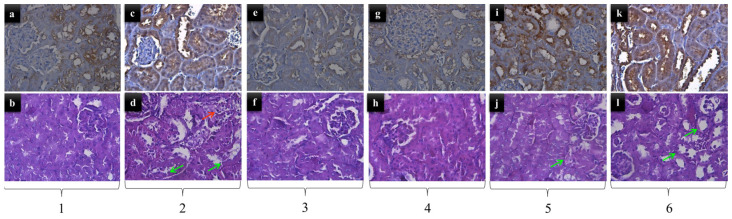
Hematoxylin eosin (HE) and GPX4 imaging. 1—control mice; 2—Gentamicin-treated mice; 3—Vitamin E-only-treated mice; 4—Silymarin-only-treated mice; 5—Gentamicin-plus-vitamin-E-treated mice; 6—Gentamicin plus silymarin-treated mice; (**a**) moderate expression of GPX4 in the kidney (×400); (**b**) normal appearance of the kidney (HE) without significant pathological changes (×400); (**c**) strong expression of GPX4 in the kidney (×400); (**d**) mild degenerative, inflammatory and vascular congestion (HE) of the kidney (×400); (**e**) weak expression of GPX4 in the kidney (×400); (**f**) normal appearance of the kidney (HE) without significant pathological changes (×400); (**g**) moderate expression of GPX4 in the kidney (×400); (**h**) normal appearance of the kidney (HE) without significant pathological changes (×400); (**i**) very strong expression of GPX4 in the kidney (×400); (**j**) very weak degeneration (HE) of the kidney (×400); (**k**) strong expression of GPX4 in the kidney (×400); (**l**) very weak degeneration (HE) of the kidney (×400). Green arrows—degeneration; red arrows—inflammation.

**Figure 2 pharmaceuticals-16-01365-f002:**
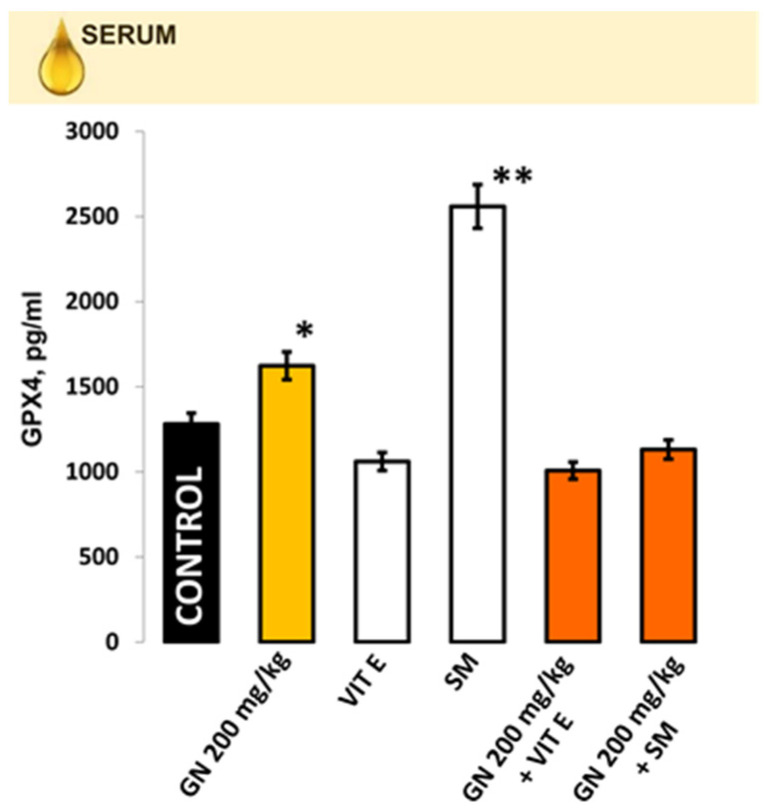
Levels of GPX4 were measured in the serum of mice (n = 6). 1—control mice; 2—Gentamicin-treated mice; 3—Vitamin E-only-treated mice; 4—Silymarin-only-treated mice; 5—Gentamicin-plus-vitamin-E-treated mice; 6—Gentamicin plus silymarin-treated mice. The quantitative data were expressed as the means ± SE. * *p* < 0.05 vs. control group; ** *p* < 0.05 vs. gentamicin-treated group.

**Figure 3 pharmaceuticals-16-01365-f003:**
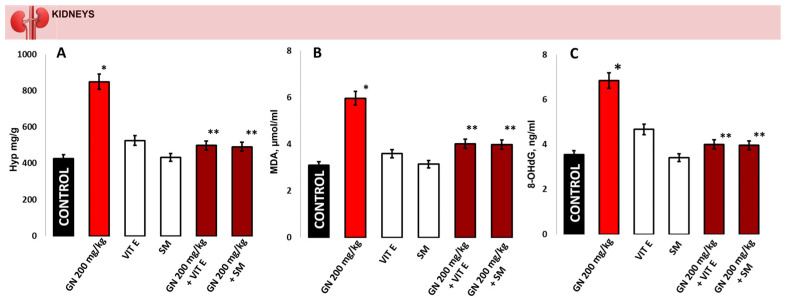
(**A**) The vitamin E and silymarin effects on GN-induced oxidative changes in kidney hydroxyproline content (mg/g); (**B**)—lipid peroxidation measured as MDA (μmol/mL); and (**C**)—DNA oxidation measured as 8-OHdG (ng/mL). The results are presented as means ± SE. * *p* < 0.05. The significant difference was used in relation to the controls (*) *p* < 0.05 vs. controls, (**) *p* < 0.05 vs. GN-treated mice.

**Figure 4 pharmaceuticals-16-01365-f004:**
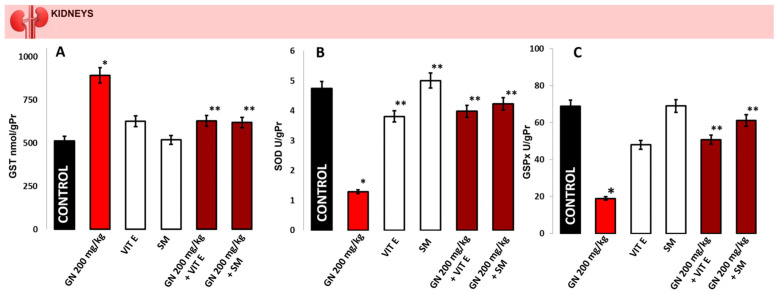
The activity of antioxidant enzymes GST (**A**), SOD (**B**), and GSPx (**C**) was presented in control mice, mice treated with GN, vitamin E, silymarin, and a combination of vitamin E + GN and silymarin + GN. The presented data means ± SE. * *p* < 0.05. (*) indicate significant differences from (* *p* < 0.05 vs. control, ** *p* < 0.05 vs. GN). No significant changes in the activity of GST, SOD and GSPx were indicated between the mice treated with antioxidants alone and the control group.

**Figure 5 pharmaceuticals-16-01365-f005:**
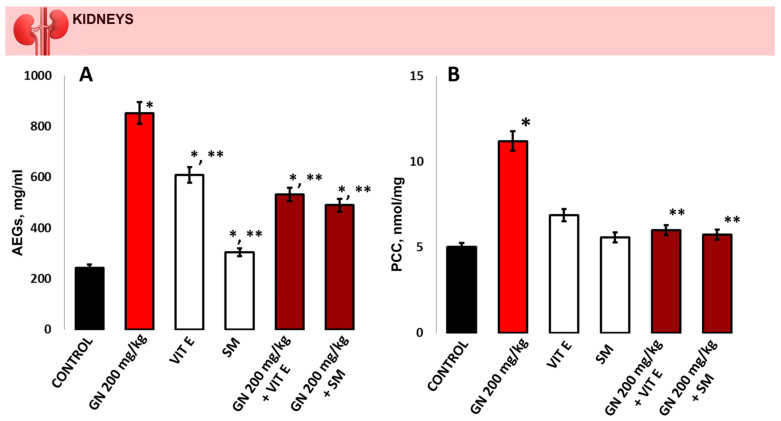
(**A**) AGEs in controls, mice treated with GN, vitamin E, silymarin, and a combination of vitamin E + GN and silymarin + GN, *p* < 0.05; (**B**) 5-MSL in the kidney of control mice, mice treated with GN, vitamin E, silymarin, and a combination of vitamin E + GN and silymarin + GN. The presented data as means ± SE. * *p* < 0.05. (*) indicate significant differences from (* *p* < 0.05 vs. control, ** *p* < 0.05 vs. GN).

**Figure 6 pharmaceuticals-16-01365-f006:**
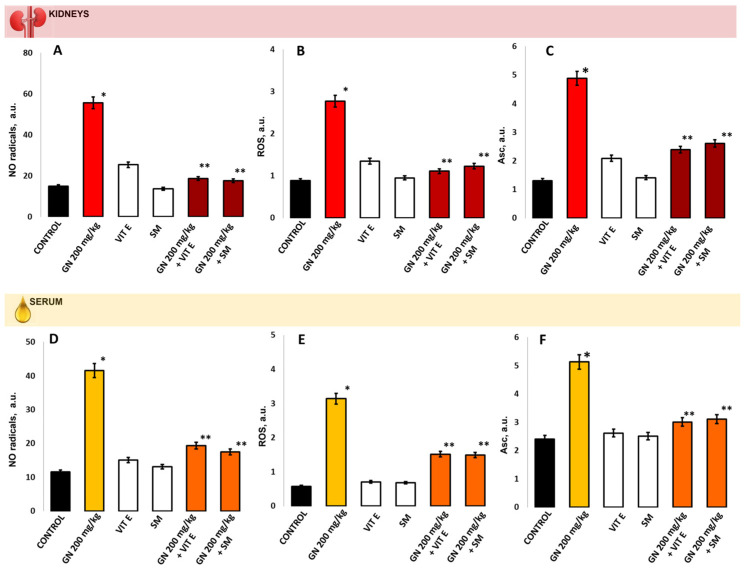
(**A**,**D**) NO radicals measured in arbitrary units (a.u.) in the kidney/serum of control mice, mice treated with GN, vitamin E, silymarin, and a combination of vitamin E + GN and silymarin + GN; (**B**,**E**) ROS (a.u.) measured in the kidney/serum of control mice, mice treated with GN, vitamin E, silymarin, and a combination of vitamin E + GN and silymarin + GN; (**C**,**F**). Asc radicals measured in arbitrary units (A.U.) in the kidney/serum of control mice, mice treated with GN, vitamin E, silymarin, and a combination of vitamin E + GN and silymarin + GN; The presented data as means ± SE. * *p* < 0.05. (*) indicate significant differences from (* *p* < 0.05 vs. control, ** *p* < 0.05 vs. GN).

**Figure 7 pharmaceuticals-16-01365-f007:**
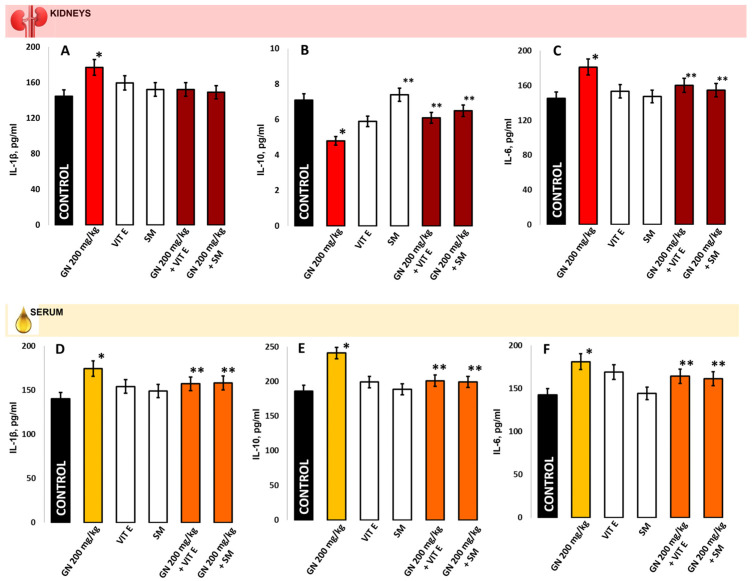
The expressions of IL-1β (**A**,**D**), IL-10 (**B**,**E**), IL-6 (**C**,**F**) in mouse kidney homogenate/serum samples; The presented data as means ± SE. * *p* < 0.05. (*) indicate significant differences from (* *p* < 0.05 vs. control, ** *p* < 0.05 vs. GN).

**Figure 8 pharmaceuticals-16-01365-f008:**
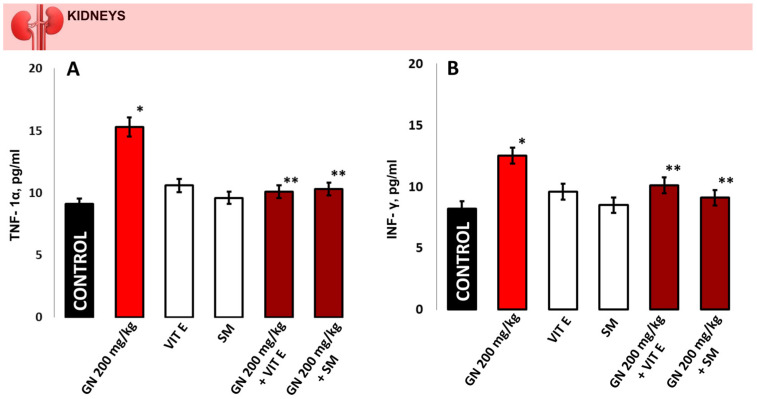
The concentration of TNF-α (**A**,**C**) and INF-γ (**B**,**D**) in the mouse kidney homogenate/serum samples; The presented data as means ± SE. * *p* < 0.05. (*) indicate significant differences from (* *p* < 0.05 vs. control, ** *p* < 0.05 vs. GN).

**Figure 9 pharmaceuticals-16-01365-f009:**
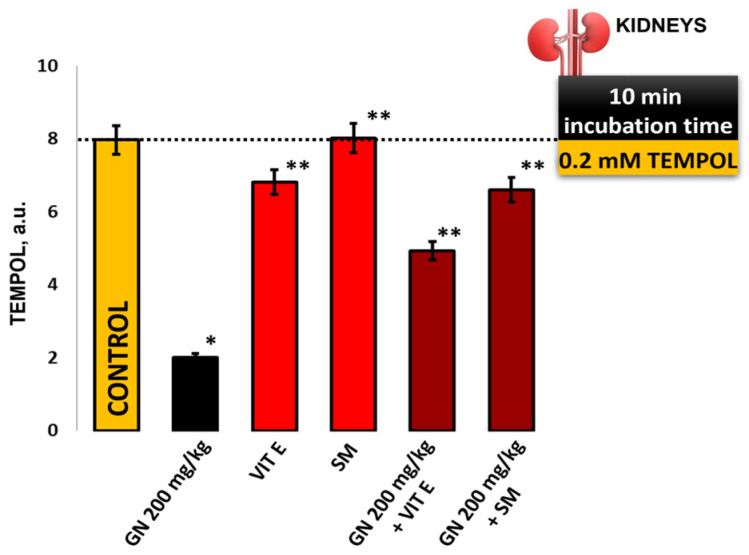
Double integration of TEMPOL spectra in mouse kidney samples, the results are presented as arbitrary units (a.u.) relative to the TEMPOL/DMSO control; The presented data as means ± SE. * *p* < 0.05 (*) indicate significant differences from (* *p* < 0.05 vs. control, ** *p* < 0.05 vs. GN).

**Table 1 pharmaceuticals-16-01365-t001:** Biochemistry parameters: creatinine, urea, and electrolytes Na^+^ and K^+^ examined at the end of the experimental period were without significant differences between all the groups (n = 6). The quantitative data were expressed as the means ± SE. * *p* < 0.05 vs. control group; ** *p* < 0.05 vs. gentamicin-treated group.

Administration (n = 6)	Creatinine Umol/L	Urea mmol/L	Na^+^ mmol/L	K^+^ mmol/L
Control	27.2 ± 3.2	7.5 ± 1.3	148.5 ± 5.1	5.3 ± 0.7
GN 200 mg/kg	37.1 ± 3.3 *	12.6 ± 2.3 *	142.4 ± 4.8	6.7 ± 1.2
VIT E	27.3 ± 2.8	6.8 ± 1.1	152.0 ± 5.2	5.4 ± 0.6
SM	26.0 ± 2.6	5.7 ± 0.9	149.1 ± 5.4	6.2 ± 0.8
GN 200 mg/kg + VIT E	22.1 ± 2.3	8.7 ± 1.3 **	151.2 ± 4.9	5.9 ± 0.8
GN 200 mg/kg + SM	24.2 ± 2.5	8.9 ± 1.3 **	149.3 ± 4.5	6.2 ± 1.1

**Table 2 pharmaceuticals-16-01365-t002:** Comparative pathomorphological changes in six groups: Control; Gentamicin-treated mice; Vitamin E-only-treated mice; Silymarin-only-treated mice; Gentamicin-plus-vitamin-E-treated mice; Gentamicin plus silymarin-treated mice.

Groups (n = 6)	GPX4	Degeneration	Necrosis	Inflammation	Hyperemia
Control	3+	0	0	0	0
GN 200 mg/kg	2+	1	0	1	1
VIT E	1+	0	0	0	0
SM	2+	0	0	0	0
GN 200 mg/kg + VIT E	4+	0/1	0	0	0
GN 200 mg/kg + SM	3+	0/1	0	0	1

Legend: 0—No changes; 1—Weak changes; 1+—Weak expression; 2+—Moderate expression; 3+—Strong expression; 4+—Very strong expression.

## Data Availability

Data is contained within the article.

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
