# Peer review of "Vitamin E and Silymarin Reduce Oxidative Tissue Damage during Gentamycin-Induced Nephrotoxicity"

_pharmaceuticals, 2023, doi:10.3390/ph16101365_

Round 1
Reviewer 1 Report
In this study, Georgiev et al. investigate the physiological roles of the antioxidative substances Vitamin E and Silymarin during Gentamycin-induced nephrotoxicity. They mainly focused on the analysis of ferroptosis during this study. They provide useful in-vivo findings, not only for gentamycin-induced nephrotoxicity in general, but also for the effects of Vitamin E and Silymarin treatment. In general, the authors provide solid evidence by using a broad spectrum of techniques to analyze their in-vivo parameters.
From the experimental site this is already a solid paper.
The manuscript in general is well written and understandable. There is, however, a little work to do concerning some parts of the text and the structure of the manuscript, as well as the display of some figures (see major and minor points for the text).
If the points mentioned in detail below can be addressed by the authors in a minor revision, this article is ready for publication and will be a useful contribution to the field.

In this study, Georgiev et al. investigate the physiological roles of the antioxidative substances Vitamin E and Silymarin during Gentamycin-induced nephrotoxicity. They mainly focused on the analysis of ferroptosis during this study. They provide useful in-vivo findings, not only for gentamycin-induced nephrotoxicity in general, but also for the effects of Vitamin E and Silymarin treatment. In general, the authors provide solid evidence by using a broad spectrum of techniques to analyze their in-vivo parameters.
From the experimental site this is already a solid paper.
The manuscript in general is well written and understandable. There is, however, a little work to do concerning some parts of the text and the structure of the manuscript, as well as the display of some figures (see major and minor points for the text).
If the points mentioned in detail below can be addressed by the authors in a minor revision, this article is ready for publication and will be a useful contribution to the field.
Author Response
Dear reviewer,
Thank You for your positive, very helpful, and constructive notes.
We hope that we have answered properly to all notes correctly. We hope that the corrections made in the text are accurate and satisfy your requirements.
We are applying the list of corrections, in the manuscript the corrections are colored red:
Point 1: The title is too general. Please rephrase to something more explanatory for your manuscript like "Vitamin E and Silymarin reduce oxidative tissue damage during gentamycin-induced nephrotoxicity".
Response 1: We rephrased the title as your proposal: "Vitamin E and Silymarin reduce oxidative tissue damage during gentamycin-induced nephrotoxicity".
Point 2: Please keep the broadly accepted pattern of “Introduction, Materials & Methods, and Results”. It is much easier to understand your experimental findings when the reader reads the “Material & Methods”-Section first.
Response 2: We agree with the reviewer, but unfortunately, we have used the template of Pharmaceuticals journal, and the pattern is “Introduction, Results, Discussion, and Materials & Methods,”.
Point 3: In the “Material & Methods”-Section: Some parts need more explanation in detail. The sub-sections 4.5, 4.8, and 4.9 need to be explained in detail. Citations of the methods are important but, left alone, not sufficient.
Response 3: Done
Point 4: In the “Results”-Section: Please put your Figure legends always under the figures (like you nicely did in Figure 8) and not above them (like in Figures 1, 2, 3, 4, 5, 6, 7). It is confusing for the reader and can be easily mistaken as a normal part of the text.
Response 4: Done
Point 5: In the “Results”-Section: Please move and explain Figure 1 before Table 2 since the values of Table 2 are the analyzed data of Figure 1.
Response 5: Done
Point 6: In Figure 1: The authors mentioned the sub-sections “a-l”, however in the Figure only the numbers 1-6 for the different treatments occur.
Response 6: Done
Point 7: In Figure 8: In panels A and C, it should be TNF-α in the y-axis, not TNF-1α. In panels B and D I think the authors meant IFN-γ and not TNF-γ in the y-axis. Please correct.
Response 7: Done
Point 8: A small explanation about the EPR method (how it works and what is measured) should be added to the main text right before Figure 9 since lipid peroxidation is a crucial parameter for this manuscript and this measurement is very important for the story. The methodical explanation in the “Material & Methods”-section can be easily overlooked and comes too late in this manuscript.
Response 8: Done.
Point 9: Please explain in the text what TEMPOL as a chemical substance exactly is doing (SOD mimetic, which converts O2¯ into H2O2) and why it was chosen for the experiment.
Response 9: Done
Point 10: Some parts of the “Results”-section put the Figures nicely in a textual context (like in lines 85-105 or 138-165). However, some parts are just the sub-title and then the figure and figure legend without any explanatory text or a small interpretation of the data (like in lines 275-280). Please always explain in a text passage what experiment is coming up next, why it was performed, and what can be interpreted.
Response 10: Done
Point 11: In line 44: Please rephrase to “accumulation of ROS and subsequent lipid peroxidation as well as oxidative cell death”.
Response 11: Done
Point 12: In lines 312-314: It is the other way around. Excessive ROS overproduction (especially in the context of ferroptosis and lipid damage) leads to membrane disruption and ion imbalances. Please rephrase.
Response 12: Done
Point 13: In line 318: IL-10 is not an inflammatory, but an anti-inflammatory marker. Please explain this contrasting finding instead of just listing it.
Response 13: Done
Point 14: In line 338: In a professional manuscript (regardless of what journal it will be published in) the phrase “data not shown” is not appropriate. Every data set, that fits into the story, is interesting for the reader. So the authors should either show the data in the main figures or a supplementary figure or remove the whole text passage.
Response 14: The phrase “data not shown” and the passage associated with it are removed.

Reviewer 2 Report
Dear Authors,
Gentamicin-induced nephrotoxicity (GIN) represents complex entity with a still unclear pathogenesis. However, it is well known that this nephrotoxicity is dose-dependent and leads to functional and morphological changes in the kidney such as elevated blood urea and creatinine level in serum, declined glomerular filtration rate, edema, and acute injury in proximal tubules. Recently, pathological events leading to cell death and involving non-apoptotic pathways, such as necroptosis, pyroptosis and ferroptosis, have become increasingly of great interest.
Your manuscript presents a study interesting and significant both for its molecular and clinical aspects.
In general, title and design are appropriate, accompanied by the methodology correctly chosen and applied. Among corrections, the most serious necessary changes I have included in the attached file.
My recommendation is to accept manuscript for publication after major revision.

None
Author Response
Dear reviewer,
Thank You for your positive, very helpful, and constructive notes. We hope that we have answered properly to all notes correctly. We hope that the corrections made in the text are accurate and satisfy your requirements. We are applying the list of corrections, and in the manuscript, the corrections are highlighted in yellow:
Point 1: All abbreviations that appear for the first time in the text should be explained. (ROS)
Response 1: Done
Point 2: Results showed also significant changes of ferroptosis-associated marker, like increased expression levels of acyl-CoA synthetase long-chain family member 4.
Response 2: Done
Point 3: remove the hyphen (ferro-static)
Response 3: Done
Point 4: Scientific names of plants are treated as Latin words, regardless of their origin, and are written in italics. (Silybum marianum L.)
Response 4: Done
Point 5: the colon missing (parameters)
Response 5: Done
Point 6: Results for the same parameter should be given with the same accuracy (number of digits after the decimal point). This applies to all reported results, both in the table and in the text.
Response 6: Done
Point 7: 3 without “+”
Response 7: Done
Point 8: 2.4. Effect of to “activity” of
Response 8: Done
Point 9: Asc radical – now explained as “the ascorbate radical”
Response 9: Done
Point 10: IL-1b to IL-1 β
Response 10: Done
Point 11: What does it mean? SE or SEM? Truly, the value of SEM is always lower than SE...
Response 11: Done
Point 12: OS – now explained as “oxidative stress” first
Response 12: Done
Point 13: There are very well-documented data, proving that antioxidants, scavengers of oxygen radicals as well as repair proteins induced by oxidative stress generated by antibiotics protect bacterial cells against death thus significantly decreasing the effectiveness of antibiotic therapies used in infectious diseases treatment. It would be good to discuss the benefits and their disadvantages...
Response 13: Done
Point 14: Title formatting errors. You need to be consistent - the same level of subheadings should be italicized everywhere, or nowhere.
Response 14: Done
Point 15: Title formatting errors. You need to be consistent - the same level of subheadings should be italicized everywhere, or nowhere.
Response 15: Done
Point 16: H2O now subscribed - H2O
Response 16: Done
Point 17: The sensitivity and specificity of this method, which was published in 1961, should be clarified. There are many methods for the quantitative determination of hydroxyproline in tissues, including colorimetric methods with a sensitivity of 0.1µg. Another established simple, accurate, and sensitive method to analyze Hyp content in rat kidney tissue is HPLC with pre-column derivatization (Liu EG, Fu XC, Qian Y, Bai HB. [Determination of hydroxyproline in rat kidney tissue by HPLC with pre-column derivatization]. Zhejiang Da Xue Xue Bao Yi Xue Ban. 2012 Mar;41(2):188-91. ] (spectrometric analysis for Hyp levels).
Response 17: We have used the spectrophotometry method of Woessner, J.B. The determination of hydroxyproline in tissue and protein samples containing small proportions of this amino acid. Arch. Biochem. Biophys. 1961, 93, 440–447.
Point 18:
Why "experimental"? The aspartate aminotransferase (AST) to alanine aminotransferase (ALT) ratio (AST/ALT) is often used in the diagnosis and prognosis of liver diseases, but it is also used in the diagnosis and prognosis of many other diseases, such as myocardial infarction, acute ischemic stroke, and peripheral arterial disease. Acute kidney injury (AKI) is one of the most important complications, e.g. after cardiac surgery, and it is known that preoperative and postoperative elevated AST to ALT ratio seems to be associated with an increased incidence of AKI. There are many known biomarkers of functional kidney injury, such as cystatin C, interleukin-18 (Il-18), kidney injury molecule-1, and neutrophil gelatinase.
Response 18: Done
Point 19: Why "experimental"? The aspartate aminotransferase (AST) to alanine aminotransferase (ALT) ratio (AST/ALT) is often used in the diagnosis and prognosis of liver diseases, but it is also used in the diagnosis and prognosis of many other diseases, such as myocardial infarction, acute ischemic stroke, and peripheral arterial disease. Acute kidney injury (AKI) is one of the most important complications, e.g. after cardiac surgery, and it is known that preoperative and postoperative elevated AST to ALT ratio seems to be associated with an increased incidence of AKI.
Response 19: Done
Point 20: In fact, there are many known biomarkers of functional kidney injury, such as cystatin C, interleukin-18 (Il-18), kidney injury molecule-1, neutrophil gelatinase
Why "experimental"? The aspartate aminotransferase (AST) to alanine aminotransferase (ALT) ratio (AST/ALT) is often used in the diagnosis and prognosis of liver diseases, but it is also used in the diagnosis and prognosis of many other diseases, such as myocardial infarction, acute ischemic stroke, and peripheral arterial disease. Acute kidney injury (AKI) is one of the most important complications, e.g. after cardiac surgery, and it is known that preoperative and postoperative elevated AST to ALT ratio seems to be associated with an increased incidence of AKI.
Response 20: Done
Point 21: In fact, there are many known biomarkers of functional kidney injury, such as cystatin C, interleukin-18 (Il-18), kidney injury molecule-1, neutrophil gelatinase
Response 21: Done
Point 22: The materials and methods do not include biochemical tests (creatinine, urea, electrolytes), while the Results section does not include the results of enzyme activity tests: the aspartate aminotransferase (AST), alanine aminotransferase (ALT), and alkaline phosphatase (ALP), which was assessed. These results were not discussed.
The EPR signal obtained with this metho is indicative of the intensity of ROS, as you wrote, but to comment on lipid peroxidation, it would be necessary ex. to determine thiobarbituric acid reactive substances (TBARS) / lipid peroxidation products. So, this title is inappropriate and misleading. You write about MDA only in the subsection: enzyme immunoassays.
Response 22: Done
Point 23: The EPR signal obtained with this metho is indicative of the intensity of ROS, as you wrote, but to comment on lipid peroxidation, it would be necessary ex. to determine thiobarbituric acid reactive substances (TBARS) / lipid peroxidation products. So, this title is inappropriate and misleading. You write about MDA only in the subsection: enzyme immunoassays.
Response 23: Done
Point 24: What was the basis for using the Student's t-test? Was the normality of the distribution of results checked? and with what tests? With n = 6, it would be surprising to obtain a normal distribution. I suppose that's why the results are reported as standard error rather than standard deviation, and Figure 9 even mentions SEM.
Response 24: Done

Round 2
Reviewer 2 Report
Dear Authors,
Your work has been sufficiently improved. I noticed incorrect numbering of one of the subtitles - line 509: it is 2.6, but it should be 4.6.